# Association between complete blood-count-based inflammatory scores and hypertension in persons living with and without HIV in Zambia

**Lackson Mwape** [1]*, **Benson M. Hamooya**[1], **Emmanuel L. Luwaya** [1], **Danny Muzata**[2], **Kaole Bwalya** [1], **Chileleko Siakabanze**[1], **Agness Mushabati**[2], **Sepiso K. Masenga**[1,3]

1 Mulungushi University, School of Medicine and Health Sciences, Livingstone, Zambia, 2 Department of Disease Control, The University of Zambia, School of Veterinary Medicine, Lusaka, Zambia, 3 Vanderbilt Institute for Global Health, Vanderbilt University Medical Center, Nashville, TN, United States of America

* luckyprincemwape5@gmail.com

## Abstract

### Background

Hypertension is a risk factor for cardiovascular events. Inflammation plays an important role in the development of essential hypertension. Studies assessing the association between complete blood count-based inflammatory scores (CBCIS) and hypertension are scarce. Therefore, this study aimed to determine the relationship between CBCIS and hypertension among individuals with and without human immunodeficiency virus (HIV).

### Method

This was a cross-sectional study among 344 participants at Serenje District Hospital and Serenje Urban Clinic. We used structured questionnaires to collect sociodemographic, clinical and laboratory characteristics. CBCIS included lymphocyte-monocyte ratio (LMR), neutrophil-lymphocyte ratio (NLR), platelet-lymphocyte ratio (PLR), derived neutrophil-lymphocyte ratio (d-NLR), and differential white blood cells. The primary outcome variable was hypertension defined as systolic and diastolic blood pressure higher than or equal to 140/90 mmHg. Logistic regression was used to estimate the association between hypertension and CBCIS in statistical package for social science (SPSS) version 22.0.

### Results

The participants had a median age of 32 years (interquartile range (IQR) 24–42) and 65.1% (n = 224) were female. The prevalence of hypertension was 10.5% (n = 36). Among those with hypertension, 55.6% (n = 20) were female and 44.4% (n = 16) were male. The CBCIS significantly associated with hypertension in people living with HIV (PLWH) was PLR (adjusted odds ratio (AOR) 0.98; 95% confidence interval (CI) 0.97–0.99, p = 0.01) while in people without HIV, AMC (AOR 15.40 95%CI 3.75–63.26), ANC (AOR 1.88 95%CI 1.05–3.36), WBC (AOR 0.52 95%CI 0.31–0.87) and PLR (AOR 0.98 95%CI 0.97–0.99) were the

**Data Availability Statement:** All relevant data are within the manuscript and its Supporting Information files.

**Funding:** This work was supported by the Fogarty International Center and National Institute of Diabetes and Digestive and Kidney Diseases of the National Institutes of Health grants 2D43TW009744 (SKM), R21TW012635 (SKM) and the American Heart Association Award Number 24IVPHA1297559 https://doi.org/10.58275/AHA.24IVPHA1297559.pc.gr.193866 (SKM).

**Competing interests:** The authors have declared that no competing interests exist.

factors associated with hypertension. Compared to people without HIV, only WBC, ANC, NLR, and d-NLR were good predictors of hypertension among PLWH.

## Conclusion

Our study indicates a notable HIV-status driven association between CBCIS and hypertension, suggesting the use of CBICS as potential biomarkers for hypertension risk with substantial implications for early detection and preventive measures.

## Introduction

Hypertension is responsible for almost half of all strokes and ischemic cardiovascular events globally [1]. Over the past 15 years, it has become evident that inflammation plays a crucial role in the development of essential hypertension [2]. The physiological mechanisms contributing to hypertension development are diverse, involving endothelial cell dysfunction, renal abnormalities, and dysregulation of the central nervous system [3]. Inflammation, however, impacts various systems, contributing to the onset of hypertension [2]. Inflammation is an intricate process that encompasses various cell types and secreted factors, many of which have been associated with hypertension [4]. While inflammation is crucial in the initiation and sustenance of hypertension, the past decade has yielded novel discoveries that have advanced the field and offered new mechanistic insights [5,6]. There is increasing evidence supporting the role of the host's inflammatory response in the onset and advancement of hypertension [5]. Complete blood count (CBC)-based biomarkers serve as surrogate inflammatory indicators derived from blood cell analysis [7]. Biomarkers based on CBC, including absolute neutrophil count (ANC), absolute lymphocyte count (ALC), absolute monocyte count (AMC), absolute eosinophil count (AEC), absolute basophil count (ABC), and absolute platelet counts (APC), as well as combinations like neutrophil-lymphocyte ratio (NLR), lymphocyte-monocyte ratio (LMR), and platelet-lymphocyte ratio (PLR), have been reported to indicate both systemic and local inflammation linked to the progression and prognosis of hypertension [8,9]. The NLR, among these complete blood count-based inflammatory scores (CBCIS), is commonly utilized as a reliable biomarker for systemic inflammatory status and is frequently associated with hypertension, as demonstrated in some studies [6,8].

As per the World Health Organization (WHO), around 38.8 million individuals are currently affected by the Human Immunodeficiency Virus (HIV) as of 2021. While the impact of the epidemic differs significantly across countries and regions, the WHO African Region is the most heavily affected [10]. In this region, nearly 3.4% of adults, or approximately 1 in 25, are people living with HIV (PLWH), constituting more than two-thirds of the global population of PLWH [10]. Cardiovascular disease has become a prominent cause of death among PLWH [11]. High blood pressure is the primary cause of cardiovascular disease [3]. Hypertension is now acknowledged as a significant chronic comorbidity in PLWH and is linked to heightened morbidity and mortality [10,12]. In PLWH, hypertension is associated with an elevated risk of hospitalization and adverse cardiovascular and renal outcomes [13]. These include progression to end-stage renal disease, ultimately reducing life expectancy and contributing to the already substantial treatment costs for this population [1]. PLWH on combination antiretroviral therapy (ART) are at an increased risk of developing hypertension than HIV-uninfected individuals [11]. Global data indicates that 35% of adults with HIV on ART have hypertension, surpassing the estimated 30% prevalence in HIV-uninfected adults [13]. Many factors

contribute to the elevated prevalence of hypertension in the HIV-infected population [5]. These include older age, male gender, a family history of hypertension, longer duration of HIV infection, low CD4 count, high viral burden, and a high body mass index [13]. In addition, specific medications included in combination ART regimens have been demonstrated to elevate the risk of hypertension in PLWH [10]. In a recent meta-analysis involving over 44,000 HIV-infected patients, the risk of hypertension was found to be twice as high in patients exposed to ART compared to those who were treatment-naïve [14,15].

Zambia is experiencing a rise in hypertension cases among both PLWH and those without HIV. Recent studies on cardiovascular diseases (CVD) across the country have revealed that hypertension is a major risk factor for the development of various CVDs, including coronary artery disease, congestive heart failure, and atrial fibrillation, among others [1,16]. Various global studies have demonstrated an association between CBCIS and hypertension in both individuals with and without HIV [6,12]. Nevertheless, there have been limited data from low- and middle-income countries to explore the connection between CBCIS and hypertension. Consequently, the objective of this study was to establish the relationship between CBCIS and hypertension among adult individuals with and without HIV in two urban health facilities of Zambia.

## Methods

### Study design and site

This was a cross-sectional study that took place at Serenje Urban Clinic and Serenje District Hospital in the small district of Serenje, located in the Central Province of Zambia. The participants included PLWH attending routine antiretroviral therapy (ART) and volunteers who tested negative for HIV and were coming in for clinical check-ups. The estimated population of Serenje is approximately 148,000, according to the Central Statistics of Zambia as of 2019.

### Eligibility and recruitment

We purposively selected adult participants, both male and female, aged 18 years and above, attending routine ART clinic and volunteers who tested negative for HIV and were coming in for clinical check-ups at Serenje Urban Clinic and Serenje District Hospital. We included PLWH and those without HIV. The selection of these two facilities was deliberate, as they are in proximity and cater to communities with similar characteristics. All participants were required to sign consent forms before being included in the study. Those with other non-communicable diseases, such as diabetes and cancer, were excluded from participation. These individuals were excluded from the study to avoid confounding effects from other non-communicable disease and cancer.

### Sample size calculation

The sample size of 344 was determined using OpenEpi online software, considering a mean difference in blood pressure proportion of 5.42 between group 1 (without HIV) and group 2 (PLWH). The choice of 344 accounts for a variance of 235.32 and 332.33, along with a standard deviation of 15.34 and 18.23 for group 1 and group 2, respectively. This calculation aims to achieve a 95% confidence interval and 80% power in the statistical analysis of the study.

### Study variables

The primary outcome variable in this study was hypertension. The diagnosis of hypertension was established when systolic and diastolic blood pressure was equal to or greater than 140/90

mmHg on more than two occasions or when there was a history of antihypertensive medication usage. The independent variables encompassed demographic (age, sex, marital status, employment status, alcohol consumption and smoking) and clinical (BMI, HIV status, hematologic parameters, and hypertension) characteristics.

## Data collection

Data collection was conducted between March 1st 2023 and June 30th 2023. Prior to data collection, written informed consent was obtained from each participant. Socio-demographic and clinical data were gathered through in-depth interviews using a structured questionnaire and by reviewing medical records. The questionnaire, initially designed in English, underwent pretesting among non-participants at the study site to assess its validity. A 4 ml venous blood sample was collected from each participant for a complete blood count. The samples were processed in the hospital laboratory within two or three hours of collection. If not analyzed within this timeframe, the samples were refrigerated for 48 hours. All blood samples were analyzed within 24 hours of collection. All data on which this manuscript is reporting has been made available.

## Blood pressure and anthropometric measurement

The anthropometric data, including height without shoes and weight (in kilograms), were gathered, and the Body Mass Index (BMI) was computed by dividing the weight in kilograms by the square of the height in meters. Blood pressure was assessed using an automated blood pressure (BP) machine.

## Data analysis

Data obtained from this study was analysed using the SPSS Software Version 22.0 for Windows. Descriptive data were presented through frequency tables and graphs using descriptives statistics such as median, standard deviation, range, quartiles, skewness and kurtosis. Hematological parameters were compared between individuals with HIV and those without HIV using the independent t-test for normally distributed data and the Mann-Whitney U test for non-parametric data. Receiver operating characteristic curve (ROC) was used to assess which CBCIS is the better predictor of hypertension. Logistic regression models (both bivariate and multiple) were employed to investigate factors associated with hypertension. A p-value of less than 0.05 was considered indicative of a statistically significant difference.

## Ethical approval and consent to participant

Ethics approval for this study was obtained from the Mulungushi University School of Medicine and Health Sciences Research Ethics Committee (IRB: 00012281 FWA: 0002888) on 19th February 2023 and from Zambia National Health Research Authority on 23rd February 2023. Participants provided written consent before they were recruited into the study. To guarantee confidentiality and anonymity, project identification numbers were assigned to the participants. No identifying information was collected such that participants were not identified during the data collection process.

We used the Strengthening the Reporting of Observational Studies in Epidemiology (STROBE) guidelines for reporting (S1 Checklist).

## Results

### Comparison of sociodemographic and clinical factors between normotensive and hypertensive in the whole population

The study involved 344 participants, with females comprising the majority at 65.1% (n = 224), Table 1. The median age of all participants was 32 years (interquartile range (IQR) 24–42). The majority of participants were currently married, accounting for 72.7% (250), while unemployment was at 62.2% (n = 214). Of these, 10.5% (n = 36) had hypertension, and 89.5% (n = 308) did not have hypertension. Among all participants, 49.1% (n = 170) were PLWH, and 50.6% (n = 174) did not have HIV. Persons with hypertension were older than those without hypertension, Fig 1A. Persons with hypertension had higher clinical factors like BMI, AMC, ANC, and d-NLR while WBC, ALC, AEC, ABC, HbG, LMR, and PLR were similar between the two groups, Fig 1.

### Comparison of the CBCIS between people with and without HIV

We compared all CBCIS including demographic characteristics and found that all were comparable but AMC was lower while LMR and AEC were higher in individuals with HIV compared to those without HIV, Fig 2. Persons with HIV were older than those without HIV.

### Comparison of sociodemographic and clinical factors between normotensive and hypertensive in both people living with and without HIV

In people living with HIV, WBC, HBG, ANC, NLR, and d-NLR were significantly associated with hypertension. WBC, ANC, NLR, and d-NLR were higher in persons with hypertension

**Table 1. Comparison of sociodemographic and clinical factors between normotensive and hypertensive in the whole population.**

| Variables | | | Frequency / Median | P Value |
|---|---|---|---|---|
| | **Hypertensive (IQR/ %)** | **Normotensive (IQR/%)** | | |
| **Sex** | | | | |
| Male | 16 (44.4) | 104 (33.7) | 120 | 0.20 |
| Female | 20 (55.6) | 204 (66.2) | 224 | |
| **Marital status** | | | | |
| Married | 28 (77.7) | 222 (72.1) | 250 | 0.68 |
| Single | 6 (16.6) | 71 (23.1) | 77 | |
| Widowed | 2 (5.5) | 15 (4.9) | 17 | |
| **Employment status** | | | | |
| Formal | 13 (36.1) | 100 (32.5) | 113 | 0.51 |
| Unemployed | 20 (55.6) | 194 (63.0) | 214 | |
| Retired | 3 (8.3) | 14 (4.5) | 17 | |
| **Consumes alcohol** | | | | |
| Yes | 19 (52.8) | 194 (63.0) | 213 | 0.23 |
| No | 17 (47.2) | 114 (37.0) | 131 | |
| **Smoker** | | | | |
| Yes | 8 (22.2) | 56 (18.1) | 64 | 0.53 |
| No | 28 (77.7) | 252 (81.8) | 280 | |
| **HIV status** | | | | |
| Yes | 19 (11.2) | 151 (88.8) | 170 | 0.66 |
| No | 17 (9.8) | 157 (90.2) | 174 | |

IQR, interquartile range.

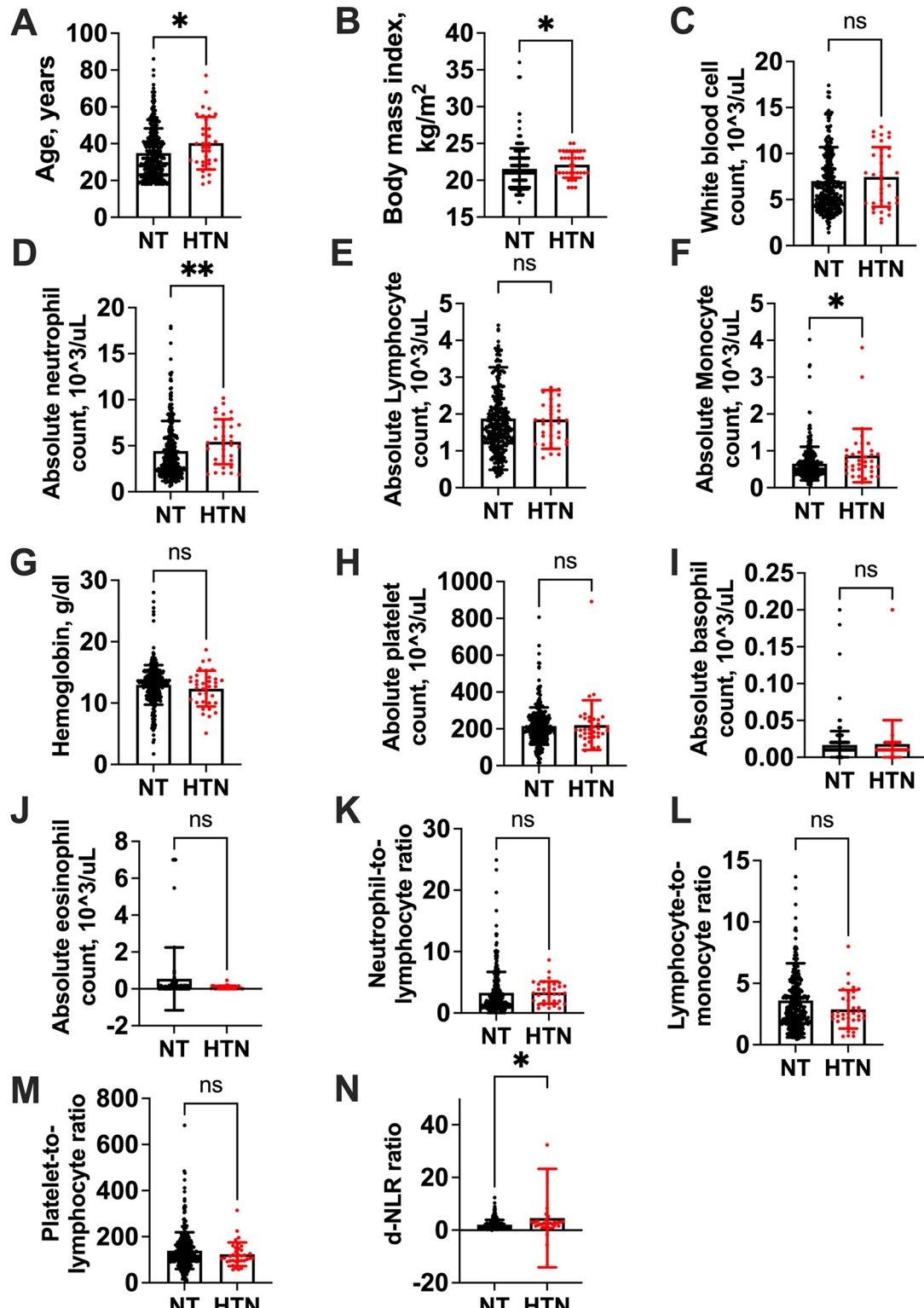

**Fig 1. Clinical and laboratory characteristics between persons with and without hypertension.** Showing median (interquartile range, IQR) between persons with and without hypertension, respectively, followed by the overall median (IQR) for the population: **A).** Age, 38 (24–42) vs. 32 (24–42) years; 32 years, p = 0.03. **B).** BMI, 22 (20.00–24.00) vs. 21 (20.00–24.00) kg/m², p = 0.04. **C).** WBC, 7.49 (4.49–8.46) vs. 6.19 (4.49–8.46) x 10³/Ul; 6.27 x 10³/Ul, p = 0.29. **D).** ANC, 5.19 (3.43–7.21) vs. 3.41 (3.43–7.21) x 10³/Ul; 3.64 x 10³/Ul, p = 0.003. **E).** ALC, 1.82 (1.21–2.15) vs. 1.63 (1.21–2.15) x10³/Ul; 1.65

x10$^3$/Ul, p = 0.39. **F).** AMC, 0.72 (0.48–1.00) vs. 0.53 (0.40–0.78) x10$^3$/Ul; 0.56 x10$^3$/Ul, p = 0.02. **G).** HbG,12.9 (11.20–14.50) vs. 13.1 (11.20–14.50) g/dl; 13.10 g/dl, p = 0.32. **H).** APC, 193.50 (158.00–251.00) vs. 202.00 (158.00–251.00) x10$^3$/Ul; 199.0 x10$^3$/Ul, p = 0.60. **I).** ABC, 0.01 (0.01–0.02) vs. 0.01 (0.01–0.02) x10$^3$/Ul; 0.01, p = 0.37. **J).** AEC, 0.04 (0.01–0.13) vs. 0.04 (0.01–0.15) x10$^3$/Ul; 0.04, p = 0.68. **K).** NLR, 3.20 (1.19–3.17) vs. 2.05 (1.19–3.17); 2.10, p = 0.06. **L).** LMR, 2.56 (2.04.-4.63) vs. 2.93 (2.04.-4.63); 2.88, p = 0.29. **M).** PLR, 111.48 (95.97–167.17) vs.122.81 (95.97–167.17); 120.63, p = 0.20. **N).** d-NLR, 2.38 (1.19–3.17) vs. 1.43 (1.19–3.17); 1.50, p = 0.03. WBC, white blood cell; AMC, absolute monocyte count; ALC, absolute lymphocyte; ANC, absolute neutrophil count; APC, absolute platelet count; HBG, hemoglobin; AEC, absolute eosinophil count; ABC, absolute basophil count; LMR, lymphocyte-monocyte ratio; NLR, neutrophil-lymphocyte ratio; PLR, platelet-lymphocyte ratio; d-NLR, derived neutrophil-lymphocyte ratio.

compared to normotensive PLWH. In contrast, among individuals without HIV, only age was significantly associated with hypertension. People with hypertension were older than normotensives, **Table 2**.

## Factors associated with hypertension in the whole population using logistic regression

On univariable logistic analysis of the whole population, the factors significantly associated with hypertension were age and AMC, but only age remained significantly associated with hypertension on multivariable analysis, **Table 3.**

## Factors associated with hypertension in people with and without HIV

We then segregated the logistic regression by HIV status and found that only PLR was negatively associated with hypertension in people with HIV, **Table 4**. In people without HIV, age, AMC and ANC were positively associated with hypertension while WBC and PLR were negatively associated with hypertension.

We performed a receiver operating characteristic curve to determine the diagnostic performance of CBCIS at all classification thresholds for hypertension, **Fig 3**. Based on the observed significant correlation, only AMC, ANC and d-NLR were significant predictors of hypertension.

## Area under the curve in both person living without HIV and person living with HIV (PLWH)

The area under the curve indicated that only d-NLR, WBC, LYM and ANC, were statistically significant predictors of hypertension in persons living with HIV while none of the CBCIS were significant predictors of hypertension in persons without HIV, **Table 5** and **Fig 4**.

## Discussion

The goal of this study was to determine the CBCIS associated with hypertension among adult individuals with and without HIV. When the two groups were combined, we found that only age was associated with hypertension. Increasing age is a known established risk factor for the development of hypertension as reported in many studies [17–19]. We then compared the CBCIS correlates of hypertension between those with and without HIV and found significant differences. In both persons with and without HIV, PLR was negatively but significantly associated with hypertension in multivariable analysis. In PLWH, PLR was the only CBCIS associated with hypertension whereas in persons without HIV WBC, AMC, ANC, and PLR were associated with hypertension suggesting that HIV-status was a significant factor determining differences in CBCIS correlates of hypertension.

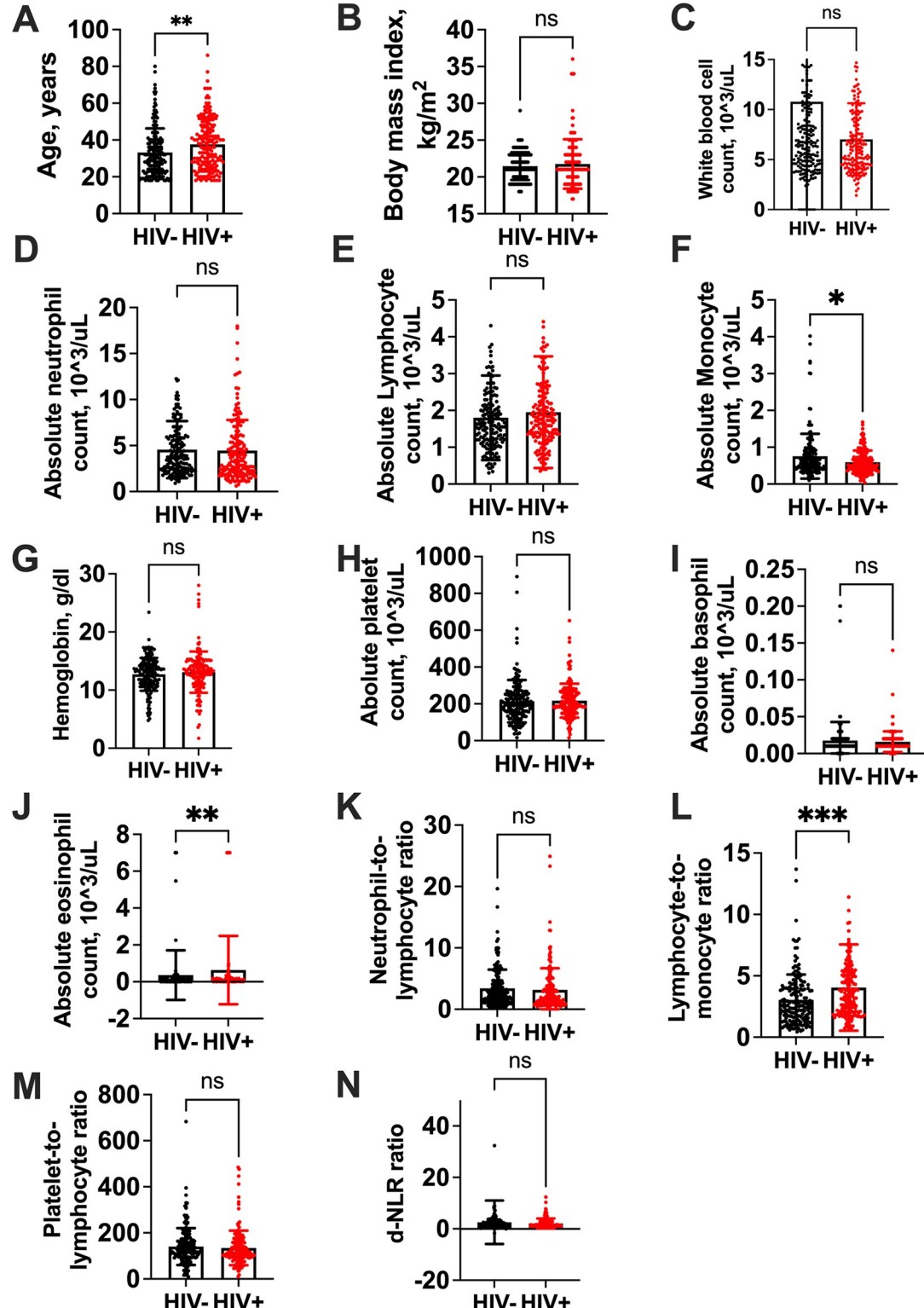

**Fig 2. Comparison of demographic and CBCIS between people with and without HIV.** Showing median (interquartile range, IQR) between persons without and with HIV, respectively: **A).** Age, 31 (23–39) vs. 36 (27–48) years, p = 0.001. **B).** BMI, 21 (20–23) vs. 21 (20–23) kg/m$^2$, p = 0.83. **C).** WBC, 6.36 (4.60–8.73) vs. 6.07 (4.49–8.97) x 10$^3$/Ul, p = 0.64. **D).** ANC, 3.74 (2.43–5.92) vs. 3.38 (2.09–5.92) x 10$^3$/Ul, p = 0.23. **E).** ALC, 1.59 (1.15–2.13) vs. 1.73 (1.29–2.25) x10$^3$/Ul, p = 0.15. **F).** AMC, 0.60 (0.43–0.86) vs. 0.50 (0.37–0.74) x10$^3$/Ul, p = 0.01. **G).** HbG,13.10 (11.10–14.40) vs. 13.20 (11.25–14.70) g/dl, p = 0.46. **H).**

APC, 199 (151–255) vs. 199 (161–264) x10$^3$/Ul, p = 0.43. **I).** ABC, 0.01 (0.01–0.02) vs. 0.01 (0.01–0.02) x10$^3$/Ul, p = 0.37. **J).** AEC, 0.03 (0.01–0.12) vs. 0.05 (0.02–0.17) x10$^3$/Ul, p = 0.004. **K).** NLR, 2.48 (1.37–4.54) vs. 1.88 (1.01–3.82), p = 0.05. **L).** LMR, 2.56 (1.66–3.96) vs. 3.31 (2.08–5.17), p<0.001 **M).** PLR, 123 (95–168) vs.115 (96–159), p = 0.31. **N).** d-NLR, 1.60 (0.94–2.96) vs. 1.35 (0.84–2.56), p = 0.22. WBC, white blood cell; AMC, absolute monocyte count; ALC, absolute lymphocyte; ANC, absolute neutrophil count; APC, absolute platelet count; HBG, hemoglobin; AEC, absolute eosinophil count; ABC, absolute basophil count; LMR, lymphocyte-monocyte ratio; NLR, neutrophil-lymphocyte ratio; PLR, platelet-lymphocyte ratio; d-NLR, derived neutrophil-lymphocyte ratio.

PLR is an emerging informative marker of inflammation relevant for predicting the course of several inflammatory conditions such as inflammatory rheumatic diseases including systemic lupus erythematosus, metabolic, neoplastic and prothrombotic diseases [20–23]. Platelet

**Table 2. Comparison of sociodemographic and clinical factors between normotensive and hypertensive in both people living with and without HIV.**

| Variables | PLWH | | P Value | Persons without HIV | | P Value |
|---|---|---|---|---|---|---|
| | Hypertensive (IQR/ %) | Normotensive (IQR/%) | | Hypertensive (IQR/ %) | Normotensive (IQR/%) | |
| **Age,** *years* | 36 (29–53) | 36 (26–47) | 0.51 | 38 (29–52) | 30 (23–38) | **0.02** |
| **Sex** | | | | | | |
| *Male* | 7 (36.8) | 41 (27.2) | 0.38 | 9 (52.9) | 63 (40.1) | 0.31 |
| *Female* | 12 (63.2) | 110 (72.8) | 0.38 | 8 (47.1) | 94 (59.9) | |
| **Marital** | | | | | | |
| *Married* | 16 (84.2) | 102 (67.5) | 0.26 | 12 (70.6) | 120 (76.4) | 0.14 |
| *Single* | 3 (15.8) | 38 (25.2) | 0.26 | 3 (17.6) | 33 (21.0) | |
| *Widowed* | 0 (0.0) | 11 (7.3) | 0.26 | 2 (11.8) | 4 (2.5) | |
| **Employment** | | | | | | |
| *Formal* | 5 (26.3) | 52 (34.4) | 0.76 | 18 (47.1) | 48 (30.6) | 0.06 |
| *Unemployed* | 13 (68.4) | 90 (59.6) | 0.76 | 7 (41.2) | 104 (66.2) | |
| *Retired* | 1 (5.3) | 9 (6.0) | 0.76 | 2 (11.8) | 5 (3.2) | |
| **Consumes alcohol** | | | | | | |
| *Yes* | 11 (57.9) | 99 (65.6) | 0.51 | 8 (47.1) | 95 (60.5) | 0.28 |
| *No* | 8 (42.1) | 52 (34.4) | 0.51 | 9 (52.9) | 62 (39.5) | |
| **Smoker** | | | | | | |
| *Yes* | 6 (31.6) | 33 (21.9) | 0.39 | 2 (11.8) | 22 (14.1) | 0.79 |
| *No* | 13 (68.4) | 118 (78.1) | 0.39 | 15 (88.2) | 134 (185.9) | |
| **WBC,** *10$^3$/Ul* | 9.0 (6.5–11.8) | 5.6 (4.4–8.4) | **0.001** | 4.6 (3.8–8.0) | 6.4 (4.6–8.7) | 0.07 |
| **AMC,** *10$^3$/Ul* | 0.61 (0.50–0.87) | 0.50 (0.37–0.72) | 0.126 | 0.90 (0.44–1.20) | 0.57 (0.43–0.83) | 0.092 |
| **ALC,** *10$^3$/Ul* | 1.8 (1.2–2.2) | 1.7 (1.3–2.4) | 0.64 | 1.8 (1.5–2.5) | 1.6 (1.1–2.1) | 0.09 |
| **ANC,** *10$^3$/Ul* | 6.0 (4.6–8.6) | 3.0 (2.0–5.2) | **<0.001** | 3.9 (2.0–5.8) | 3.7 (2.4–6.0) | 0.97 |
| **ABC,** *10$^3$/Ul* | 0.01 (0.01–0.02) | 0.01 (0.01–0.02) | 0.81 | 0.01 (0.01–0.01) | 0.01 (0.01–0.02) | 0.10 |
| **AEC,** *10$^3$/Ul* | 0.04 (0.01–0.10) | 0.05 (0.02–0.19) | 0.19 | 0.04 (0.01–0.13) | 0.03 (0.01–0.12) | 0.48 |
| **APC,** *10$^3$/Ul* | 183 153–211) | 205 (160–267) | 0.12 | 210 (159–254) | 199 (145–252) | 0.43 |
| **HbG,** *g/dl* | 12 (10.0–13.5) | 13 (11.5–14.9) | **0.01** | 13.9 (10.4–15.5) | 13.0 (11.1–14.4) | 0.32 |
| **LMR** | 2.6 (2.1–3.6) | 3.5 (2.1–5.3) | 0.11 | 2.6 (1.0–4.4) | 2.6 (1.7–4.0) | 0.90 |
| **NLR** | 3.8 (3.1–5.1) | 1.7 (0.9–3.4) | **<0.001** | 1.9 (1.2–4.0) | 2.5 (1.4–4.6) | 0.31 |
| **PLR** | 107 (95–159) | 115 (96–159) | 0.52 | 116 (93.7–144.6) | 127 (95.0–172.5) | 0.24 |
| **d-NLR** | 2.7 (3.2–3.2) | 1.2 (0.7–2.2) | **<0.001** | 1.6 (0.6–3.1) | 1.6 (1.0–3.0) | 0.52 |

WBC, white blood cell; AMC, absolute monocyte count; ALC, absolute lymphocyte; ANC, absolute neutrophil count; APC, absolute platelet count; HBG, hemoglobin; LMR, lymphocyte-monocyte ratio; NLR, neutrophil-lymphocyte ratio; PLR, platelet-lymphocyte ratio; d-NLR, derived neutrophil-lymphocyte ratio; ABC, absolute basophil count; AEC, absolute eosinophil count.

**Table 3. Factors associated with hypertension in the whole population logistic regression.**

| Variables | OR (95%CI) | P value | AOR (95% CI) | P value |
|---|---|---|---|---|
| **Age,** *years* | 1.03 (1.00–1.05) | **0.04** | 1.02 (1.00–1.05) | **0.04** |
| **Sex** *(male)* | 0.64 (0.32–1.28) | 0.21 | 0.73 (0.34–1.54) | 0.33 |
| **BMI,** *kg/m²* | 1.07 (0.95–1.20) | 0.27 | 1.08 (0.96–1.22) | 0.16 |
| **ANC** | 1.08 (0.99–1.18) | 0.09 | 1.13 (1.01–1.28) | 0.36 |
| **AMC** | 1.85 (1.10–3.13) | **0.01** | 1.66 (0.91–3.04) | 0.09 |
| **d-NLRHIV Status** *(yes)* | 1.04 (0.99–1.09)1.17 (0.59–2.34) | 0.120.66 | 1.03 (0.98–1.08)1.19 (0.59–2.41) | 0.190.67 |

BMI, body mass index; ANC, absolute neutrophil count; AMC, absolute monocyte count; d-NLR, derived neutrophil-lymphocyte ratio and HIV, human immunodeficiency virus.

index ratios in HIV are emerging markers used to evaluate immune status and HIV-related complications [24]. However, PLR must be used along other CBCIS to improve clinical utility.

WBC and specifically AMC and ANC were predictors of hypertension in persons without HIV in our study. Monocytes and neutrophils play an important role in hypertension [2,25]. AMC and ANC were positively correlated with systolic and diastolic blood pressure in a mendelian randomization study [26]. The exact mechanisms involving monocytes and neutrophils have been elucidated elsewhere [25,27,28] but were beyond the scope of our study.

Although AMC, ANC and d-NLR were the only significant predictors of hypertension in the whole population when we applied ROC curves, HIV-status abrogated this relationship.

**Table 4. Factors associated with hypertension in people with and without HIV.**

| Variables | OR (95%CI) | P value | AOR (95% CI) | P value |
|---|---|---|---|---|
| **People with HIV** | | | | |
| Age, *years* | 1.00 (0.97–1.04) | 0.59 | 1.01 (0.97–1.05) | 0.50 |
| Sex *(male)* | 1.55 (0.57–4.21) | 0.38 | 2.50 (0.64–9.67) | 0.18 |
| BMI, *kg/m²* | 1.03 (0.90–1.18) | 0.58 | 1.16 (0.97–1.40) | 0.09 |
| AMC | 1.74 (0.41–7.42) | 0.44 | 0.09 (0.00–9.30) | 0.18 |
| ANC | 1.16 (1.03–1.31) | **0.01** | 3.16 (0.82–12.09) | 0.09 |
| d-NLR | 1.20 (0.98–1.47) | 0.07 | 1.01 (0.49–2.12) | 0.95 |
| WBC | 1.13 (1.01–1.26) | **0.02** | 0.41 (0.14–1.18) | 0.10 |
| LMR | 0.75 (0.55–1.01) | 0.06 | 0.76 (0.58–1.00) | 0.09 |
| PLR | 0.99 (0.98–1.00) | 0.38 | 0.98 (0.97–0.99) | **0.01** |
| **People without HIV** | | | | |
| Age, *years* | 1.04 (1.00–1.05) | **0.01** | 1.04 (1.01–1.08) | **0.03** |
| Sex *(male)* | 1.67 (0.61–4.58) | 0.31 | 1.23 (0.40–3.78) | 0.71 |
| BMI, *kg/m²* | 1.17 (0.90–1.52) | 0.22 | 1.24 (0.91–1.68) | 0.16 |
| AMC | 1.97 (1.10–3.53) | **0.02** | 15.40 (3.75–63.26) | **<0.01** |
| ANC | 0.96 (0.80–1.15) | 0.69 | 1.88 (1.05–3.36) | **0.03** |
| d-NLR | 1.03 (0.98–1.07) | 0.13 | 1.01 (0.97–1.06) | 0.43 |
| WBC | 0.86 (0.71–1.05) | 0.15 | 0.52 (0.31–0.87) | **0.01** |
| LMR | 0.97 (0.75–1.25) | 0.86 | 0.93 (0.65–1.34) | 0.71 |
| PLR | 0.99 (0.98–1.00) | 0.37 | 0.98 (0.97–0.99) | **0.02** |

BMI, body mass index; AMC, absolute monocyte count; ANC, absolute neutrophil count; d-NLR, derived neutrophil-lymphocyte ratio; PLR, platelet-lymphocyte ratio; WBC, white blood cells and HIV, human immunodeficiency virus.

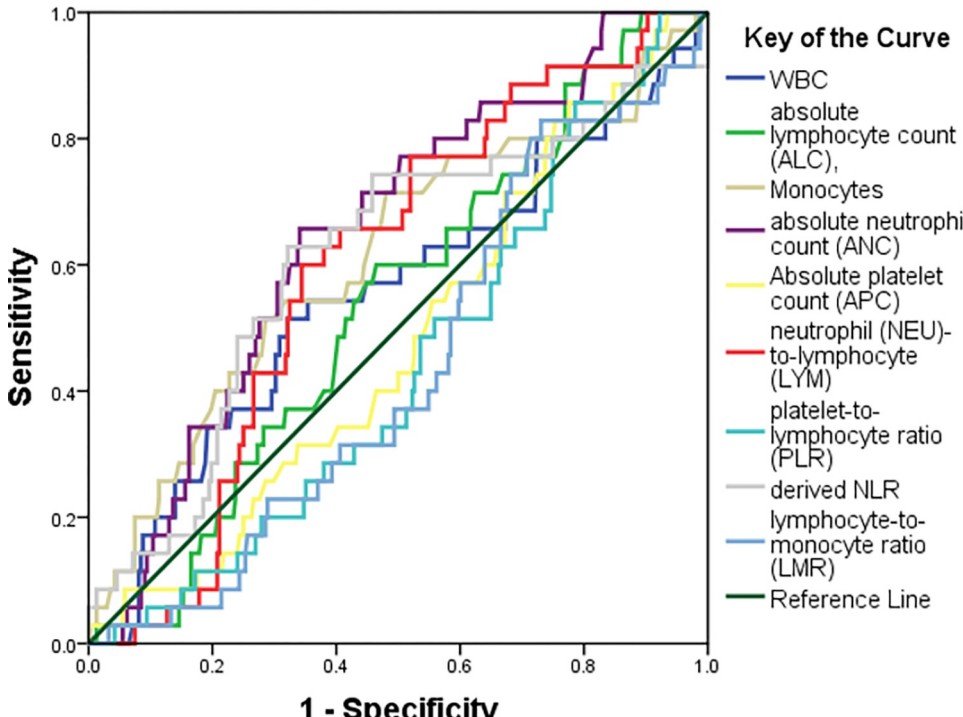

**Fig 3. ROC curve for predictors of hypertension in the whole population.** Use of the receiver operating characteristic curve to see which CBCIS score best predicts hypertension in the whole population. The area under the curve (95%CI, p = value) for WBC = 0.55 (0.44–0.66, p = 0.30), ALC = 0.53 (95%CI 0.44–0.62, p = 0.45), AMC = 0.61 (95%CI 0.50–0.71, **p = 0.029**), ANC = 0.65 (95%CI 0.56–0.73, **p = 0.004**), APC = 0.47 (95%CI 0.38–0.57, p = 0.67), NLR = 0.59 (95%CI 0.51–0.68, p = 0.05), PLR = 0.44 (95%CI 0.35–0.53, p = 0.26), d-NLR = 0.61 (95%CI 0.50–0.71, **p = 0.032**) and LMR = 0.43 (95%CI 0.34–0.52). WBC, white blood cell; AMC, absolute monocyte count; ALC, absolute lymphocyte; ANC, absolute neutrophil count; APC, absolute platelet count; HBG, hemoglobin; LMR, lymphocyte-monocyte ratio; NLR, neutrophil-lymphocyte ratio; PLR, platelet-lymphocyte ratio; d-NLR, derived neutrophil-lymphocyte ratio.

**Table 5. Area under the curve in persons living without and with HIV.**

| Table 5: | Area under the Curve in persons living without HIV | | | | | Area under the Curve in PLWH | | | | |
|---|---|---|---|---|---|---|---|---|---|---|
| Variable | Area | Std. Error | P Value | 95% Confidence interval | | Area | Std. Error | p value | 95% Confidence Interval | |
| | | | | Lower Bound | Upper Bound | | | | Lower Bound | Upper Bound |
| WBC | 0.37 | 0.077 | 0.078 | 0.21 | 0.52 | 0.73 | 0.057 | **0.001** | 0.61 | 0.84 |
| ALC | 0.63 | 0.059 | 0.077 | 0.51 | 0.74 | 0.45 | 0.065 | 0.494 | 0.32 | 0.57 |
| AMC | 0.62 | 0.089 | 0.092 | 0.45 | 0.79 | 0.61 | 0.064 | 0.129 | 0.48 | 0.73 |
| ANC | 0.50 | 0.070 | 0.982 | 0.36 | 0.63 | 0.77 | 0.045 | **<0.001** | 0.69 | 0.86 |
| APC | 0.56 | 0.070 | 0.413 | 0.42 | 0.69 | 0.39 | 0.064 | 0.156 | 0.27 | 0.52 |
| NLR | 0.42 | 0.065 | 0.287 | 0.29 | 0.54 | 0.76 | 0.039 | **<0.001** | 0.68 | 0.83 |
| PLR | 0.40 | 0.061 | 0.215 | 0.28 | 0.52 | 0.47 | 0.069 | 0.704 | 0.33 | 0.60 |
| d-NLR | 0.45 | 0.090 | 0.498 | 0.27 | 0.62 | 0.77 | 0.042 | **<0.001** | 0.69 | 0.85 |
| LMR | 0.49 | 0.083 | 0.899 | 0.32 | 0.65 | 0.37 | 0.050 | 0.075 | 0.27 | 0.46 |

WBC, white blood cell; AMC, absolute monocyte count; ALC, absolute lymphocyte; ANC, absolute neutrophil count; APC, absolute platelet count; HBG, hemoglobin; LMR, lymphocyte-monocyte ratio; NLR, neutrophil-lymphocyte ratio; PLR, platelet-lymphocyte ratio; d-NLR, derived neutrophil-lymphocyte ratio.

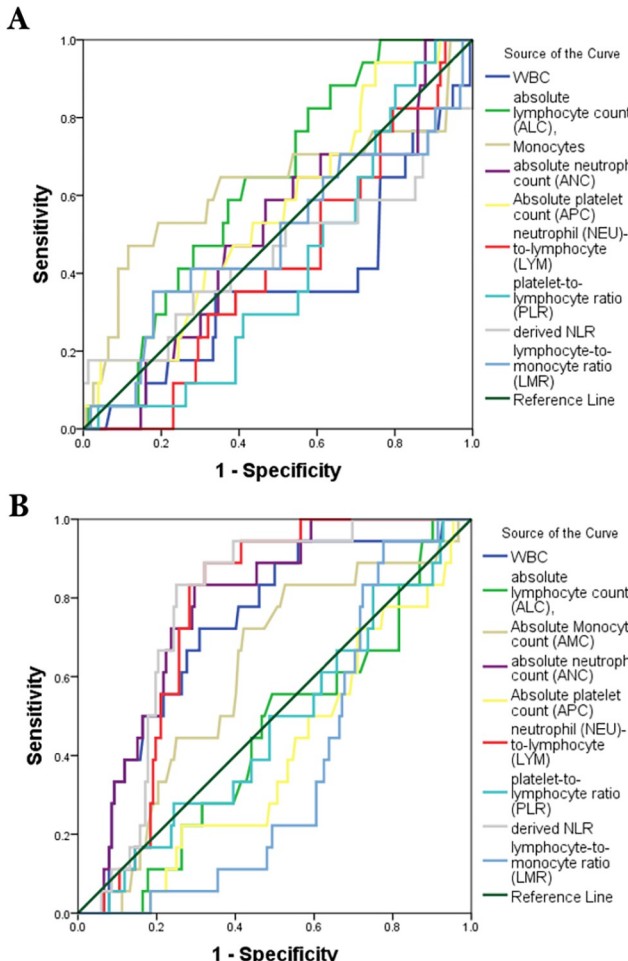

**Fig 4. ROC curve for predictors of hypertension in persons with and without HIV.** A) Persons without HIV. B). Persons with HIV. WBC, white blood cell; AMC, absolute monocyte count; ALC, absolute lymphocyte; ANC, absolute neutrophil count; APC, absolute platelet count; HBG, hemoglobin; LMR, lymphocyte-monocyte ratio; NLR, neutrophil-lymphocyte ratio; PLR, platelet-lymphocyte ratio; d-NLR, derived neutrophil-lymphocyte ratio.

None of the CBCIS predicted hypertension in persons without HIV while in PLWH, WBC, ANC, NLR, and d-NLR were significant predictors of hypertension. This makes more sense especially that HIV elicits an inflammatory response and CBCIS are expected to be different in PLWH when compared to HIV negative individuals. In PLWH, WBCs and inflammatory neutrophils and monocytes are increased due to continued subclinical inflammation even with HIV RNA viral suppression [9,29,30]. The finding that NLR as well as d-NLR were good predictors of hypertension has been confirmed from previous studies [31,32] and beyond this, NLR predicts all-cause mortality in persons with hypertension [33–35]. Elevated d-NLR is associated with poor prognosis in certain disease such as cancer [36,37] but its clinical utility in hypertension remain unknown.

In clinical settings, increasing values of CBCIS predicts worsening inflammatory disease and could therefore be used to monitor immune status and management of PLWH.

## Study limitations and strengths

Our study had some limitations and strengths. Firstly, the role of inflammation in hypertension development is crucial. Although we identified a significant association between CBCIS

and hypertension in the multivariable analysis, the lack of data on specific inflammatory markers such as acute phase proteins and hypertension-specific inflammatory cytokines was a limitation. This absence of data prevented a comprehensive understanding of whether the association between CBCIS and hypertension might be mediated by inflammation. Secondly, we did not exclude bacteremia, which could have affected some hematological parameters. Finally, the study lacked dietary information and other additional hypertension-related risk factors, preventing the determination of their influence on hypertension development. Although it is a cross-sectional study, the strength of our study lies in usage of multiple CBCIS to determine their association with hypertension. We have also included a control group of HIV negative persons to help us understand changes in CBCIS that can be explained by HIV status. To our knowledge, this is the first study to elucidate this. The regular monitoring of complete blood count parameters in persons with hypertension at our healthcare facilities can be employed to track the prognosis of hypertension and address complications at an early stage. Hence, it is essential to enhance public awareness and interventional programs targeted at managing hypertension in low cost settings. We propose for a follow up study to validate our findings before they can be applied in clinical settings.

## Conclusion

Our study indicates a notable HIV-status driven association between CBCIS and hypertension, suggesting the use of CBICS as potential biomarkers for hypertension risk with substantial implications for early detection and preventive measures.

## Supporting information

**S1 Checklist. STROBE statement—checklist of items that should be included in reports of *cross-sectional studies.***
(DOCX)

**S1 Dataset.**
(XLSX)

## Acknowledgments

The authors express gratitude to the management of the Serenje District Health Office and Serenje Urban Clinic for granting permission to collect blood samples. Appreciation is extended to the management of Serenje District Hospital in Zambia for permitting the conduction of experiments in the hospital laboratory. Special acknowledgment is also given to the Medical laboratory technologist who contributed to the data collection process.

## Author Contributions

**Conceptualization:** Lackson Mwape, Sepiso K. Masenga.

**Data curation:** Lackson Mwape.

**Formal analysis:** Lackson Mwape, Danny Muzata.

**Investigation:** Lackson Mwape.

**Methodology:** Lackson Mwape.

**Supervision:** Sepiso K. Masenga.

**Validation:** Lackson Mwape, Benson M. Hamooya, Emmanuel L. Luwaya, Danny Muzata, Kaole Bwalya, Chileleko Siakabanze, Agness Mushabati, Sepiso K. Masenga.

**Visualization:** Lackson Mwape, Benson M. Hamooya, Emmanuel L. Luwaya, Danny Muzata, Kaole Bwalya, Chileleko Siakabanze, Agness Mushabati, Sepiso K. Masenga.

**Writing – original draft:** Lackson Mwape.

**Writing – review & editing:** Lackson Mwape, Benson M. Hamooya, Emmanuel L. Luwaya, Danny Muzata, Kaole Bwalya, Chileleko Siakabanze, Agness Mushabati, Sepiso K. Masenga.

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
