## [Decision Letter · Decision Letter 0]

15 Jul 2024

PONE-D-24-12530Association between Complete Blood-Count-Based Inflammatory Scores and Hypertension in Persons Living with and Without HIV in Zambia: A cross-sectional study.PLOS ONE

Dear Dr. Mwape,

Thank you for submitting your manuscript to PLOS ONE. After careful consideration, we feel that it has merit but does not fully meet PLOS ONE’s publication criteria as it currently stands. Therefore, we invite you to submit a revised version of the manuscript that addresses the points raised during the review process.

We look forward to receiving your revised manuscript.

Kind regards,

Aleksandra Klisic

Academic Editor

PLOS ONE

Journal Requirements:

Reviewers' comments:

Reviewer's Responses to Questions

**Comments to the Author**

1. Is the manuscript technically sound, and do the data support the conclusions?

Reviewer #1: Yes

Reviewer #2: Yes

2. Has the statistical analysis been performed appropriately and rigorously? 

Reviewer #1: I Don't Know

Reviewer #2: Yes

3. Have the authors made all data underlying the findings in their manuscript fully available?

Reviewer #1: Yes

Reviewer #2: Yes

4. Is the manuscript presented in an intelligible fashion and written in standard English?

Reviewer #1: Yes

Reviewer #2: Yes

5. Review Comments to the Author

Reviewer #1: The study titled "Association between Complete Blood-Count-Based Inflammatory Scores and Hypertension in Persons Living with and Without HIV in Zambia: A cross-sectional study" by Mwape and colleagues evaluated the association between inflammatory scores and HTN in HIV and non-HIV individuals. I have some comments for improvement:

1- Define abbreviations in their first use and make sure that abbreviated forms are being used after the definition.

2- Add the clinical utility of your findings for a primary care physician.

3- I found some typos and grammatical errors. A native review is warranted.

Reviewer #2: Mwape et al. investigated the association between complete blood count-based inflammatory scores and hypertension in individuals living with and without HIV in Zambia. Although the findings are not novel, the geospatial focus is of significant interest. The study is well-contextualized and methodologically sound. However, the authors should provide a stronger emphasis on the clinical implications of their findings, which are currently not clearly articulated.

6. PLOS authors have the option to publish the peer review history of their article (what does this mean?). If published, this will include your full peer review and any attached files.

Reviewer #1: No

Reviewer #2: No

---

## [Author Response · Author response to Decision Letter 0]

7 Sep 2024

RE: RESPONSE TO REVIEWERS COMMENTS

Thank you so much for reviewing our manuscript and providing invaluable reviews. We appreciate your review comments which we believe have improved our manuscript's readability and interest.

We have taken time to correct our manuscript and provided a Track changes copy to reflect all changes in the manuscript. We have made substantial revisions in addition to what the reviewers requested including converting data with continuous variables from table 1 and 2 into figures to improve and enhance interest.

Below are our responses to the editor and reviewer comments.

EDITOR COMMENTS 

Response from authors: Thank you. We have referred to the comments and have since adjusted the naming of the manuscript and supporting documents to PLOS ONE’s style requirements.

2. PLOS require an ORCID iD for the corresponding author in Editor Manager on paper submitted after December 6th 2016. Please ensure that you have an ORCID iD and that it is validated in Editorial manager. To do this, go to ‘Update my information’ (in the upper left-hand corner of the main menu), and click on the Fetch/Validate link next to the ORCID field. This will take you to the ORCID site and allow you to create a new iD or authenticate a pre-existing iD in Editorial Manager. Please see the following video for instructions on linking an ORCID iD to your Editorial Manager account: https://www.youtube.com/watch?v=_xcclfuvtxQ

Response from authors: Thank you. We have managed to get an ORCID iD 

3. Please include captions for your Supporting Information files at the end of your manuscript, and update any in-text citations to match accordingly. Please see our Supporting Information guidelines for more information:http://journals.plos.org/plosone/s/supporting-information.

Response from authors: Thank you, we have included the captions as per recommendation.

RESPONSES TO REVIEWER COMMENTS 

REVIEWER 1

1. Define abbreviation in their first use and make sure that abbreviated forms are being used after definitions.

Response from authors: Thank you very much. We have made the revisions and defined all abbreviations upon first use, thereafter, we used the abbreviated form.

2. Add the clinical utility of your findings for a primary care physician?

Response from authors: Thank you so much. We have revised and included the clinical utility and implications of our findings in discussion section.

3. I found some typos and grammatical errors. A native review is warranted

Response from authors: Thank you so much for the comment. We have revised all the typos and grammatical errors were corrected as advised.

REVIEWER 2

1. The authors should provide a stronger emphasis on the clinical implications of their findings, which are currently not clearly articulated

Response from author: Thank you so much for the comment. We have clearly articulated and emphasized the clinical implications of our findings in the discussion section.

We want to thank the reviewers again for taking time to review and make suggestions that have improved our manuscript. We now hope it is acceptable for publication.

---

## [Decision Letter · Decision Letter 1]

26 Oct 2024

Association between Complete Blood-Count-Based Inflammatory Scores and Hypertension in Persons Living with and Without HIV in Zambia

PONE-D-24-12530R1

Dear Dr. Mwape,

We’re pleased to inform you that your manuscript has been judged scientifically suitable for publication and will be formally accepted for publication once it meets all outstanding technical requirements.

Kind regards,

Aleksandra Klisic

Academic Editor

PLOS ONE

Additional Editor Comments (optional):

Reviewers' comments:

Reviewer's Responses to Questions

**Comments to the Author**

1. If the authors have adequately addressed your comments raised in a previous round of review and you feel that this manuscript is now acceptable for publication, you may indicate that here to bypass the “Comments to the Author” section, enter your conflict of interest statement in the “Confidential to Editor” section, and submit your "Accept" recommendation.

Reviewer #1: All comments have been addressed

Reviewer #2: All comments have been addressed

2. Is the manuscript technically sound, and do the data support the conclusions?

Reviewer #1: (No Response)

Reviewer #2: Yes

3. Has the statistical analysis been performed appropriately and rigorously? 

Reviewer #1: (No Response)

Reviewer #2: Yes

4. Have the authors made all data underlying the findings in their manuscript fully available?

Reviewer #1: (No Response)

Reviewer #2: Yes

5. Is the manuscript presented in an intelligible fashion and written in standard English?

Reviewer #1: (No Response)

Reviewer #2: Yes

6. Review Comments to the Author

Reviewer #1: (No Response)

Reviewer #2: I would like to thank the authors for considering my reccomendations and appropriately revising their manuscript

7. PLOS authors have the option to publish the peer review history of their article (what does this mean?). If published, this will include your full peer review and any attached files.

Reviewer #1: No

Reviewer #2: No

---

## [Editor Report · Acceptance letter]

31 Oct 2024

PONE-D-24-12530R1 

PLOS ONE

Dear Dr. Mwape, 

I'm pleased to inform you that your manuscript has been deemed suitable for publication in PLOS ONE. Congratulations! Your manuscript is now being handed over to our production team.

Kind regards, 

on behalf of

Dr. Aleksandra Klisic 

Academic Editor

PLOS ONE